# Ionic Liquids as Performance Ingredients in Space Lubricants

**DOI:** 10.3390/molecules26041013

**Published:** 2021-02-14

**Authors:** Erik Nyberg, Christoph Schneidhofer, Lucia Pisarova, Nicole Dörr, Ichiro Minami

**Affiliations:** 1Department of Engineering Sciences and Mathematics, Division of Machine Elements, Luleå University of Technology, SE-97187 Luleå, Sweden; ichiro.minami@ltu.se; 2AC2T Research GmbH, AT-2700 Wiener Neustadt, Austria; Christoph.Schneidhofer@ac2t.at (C.S.); Lucia.Pisarova@ac2t.at (L.P.); Nicole.Doerr@ac2t.at (N.D.)

**Keywords:** space-tribology, tribofilm, silicate boundary film, multiply alkylated cyclopentanes (MAC), anti-wear, friction modifier, lubricant additive, conductivity improver

## Abstract

Low vapor pressure and several other outstanding properties make room-temperature ionic liquids attractive candidates as lubricants for machine elements in space applications. Ensuring sufficient liquid lubricant supply under space conditions is challenging, and consequently, such tribological systems may operate in boundary lubrication conditions. Under such circumstances, effective lubrication requires the formation of adsorbed or chemically reacted boundary films to prevent excessive friction and wear. In this work, we evaluated hydrocarbon-mimicking ionic liquids, designated P-SiSO, as performance ingredients in multiply alkylated cyclopentane (MAC). The tribological properties under vacuum or various atmospheres (air, nitrogen, carbon dioxide) were measured and analyzed. Thermal vacuum outgassing and electric conductivity were meas- ured to evaluate ‘MAC & P-SiSO’ compatibility to the space environment, including the secondary effects of radiation. Heritage space lubricants—MAC and perfluoroalkyl polyethers (PFPE)—were employed as references. The results corroborate the beneficial lubricating performance of incorporating P-SiSO in MAC, under vacuum as well as under various atmospheres, and demonstrates the feasibility for use as a multifunctional additive in hydrocarbon base oils, for use in space exploration applications.

## 1. Introduction

On Earth, machine designers are accustomed to the access of a wide range of liquid lubricants that enable predictable and reliable long-term operations of high-performance machinery. In space applications on the other hand, engineers are constrained to a comparatively limited choice of lubricant candidates that can meet the stringent demands of tribosystems operating in a space environment. In this paper we discuss challenges and opportunities relevant to ionic liquids in space tribology, and based on experimental evidence we provide analysis of how ionic liquids used as multifunctional additives can contribute in meeting some of those lubricant challenges.

### 1.1. Ionic Liquid Lubricants 

Ionic liquids (ILs) are interesting lubricant candidates for machine elements in space. These fluids are composed entirely of cations and anions, particles of opposite charge that display strong polarity and form ionic bonds. Despite these strong bonds, ILs do not solidify until well below room temperature—they remain mobile because of large and irregular shapes that sterically hinders crystallization. At the other end of the temperature spectrum, the strong ionic bonds reduce the fluid rate of evaporation to provide inherently low fluid vapor pressure. In total, strong ionic bonds and steric effects provide fluidity over a wide temperature range. The IL polarity also enables affinity and reactivity to a wide range of engineering materials. The strength of the ionic interaction affects viscosity and other fluid properties, and can be tuned by adjusting the size and shape of the ions. Furthermore, functional groups can be incorporated to provide task-specific fluid characteristics. Ionic liquids have attracted great research interest over the past two decades because of these unusual, attractive, and tunable properties, with the majority of research focusing on ILs as non-volatile solvents for catalysis [1]. As many of the properties that are attractive for solvents are also attractive for lubricants, tribologists have suggested the use of ionic liquids as lubricants since 2001 [2]. The concept of ILs as lubricants has been reviewed thoroughly over the years [3,4,5,6,7,8,9,10,11,12], and ILs as space lubricant have been an active field of research since the start of IL lubricant research [2,13,14,15,16,17,18,19,20,21]. Although ILs have gained a strong foothold as lubricant additives in terrestrial applications [22], their potential as space grade lubricants has not been truly realized yet [18,19]. However, considering the IL designs which have been evaluated as space grade lubricants so far, it is clear that they were originally designed for other purposes and were thus not tailored towards the specific requirements of space-grade lubricants [23,24,25]. The vast possibilities of molecular design to tailor ILs as space grade lubricants remains largely unexplored. 

### 1.2. Outlook on Space-Grade Lubricants

As international space agencies are converging on the goal of establishing a permanently crewed lunar Gateway for human presence on the Moon and eventually on Mars [26,27], there is need for radical improvements in many aspects of space exploration technology, including space tribology [28,29,30]. As described in Phase 1 and 2 of the international space exploration coordination group’s (ISECG) 2020 global exploration roadmap, landed utility vehicles will require operational capabilities that are far beyond the current state-of-the art for roving vehicles within the end of the decade [27]. Lubricants are key performance enablers in the mechanical power transmissions of these vehicles, but the current lubricant technology is already pushed to the limit to meet the demands of existing transmissions [31]. Current rover mobility rely on geared actuators that require extensive preheating to reach their minimum acceptable operating temperatures before use [32,33,34], and once in operation the lubricant supply must be ensured as lubricant starvation may lead to high wear or seizure [35].

Meanwhile in the Earth orbiting space segment, the number of satellite launches are increasing rapidly to meet the growing demand for satellites for Earth observation, positioning systems, and communications [36]. These Earth-orbiting satellites often rely on electro-mechanical momentum wheels (also commonly referred to as reaction wheels) or control moment gyroscopes for satellite attitude control. Failures in these wheels and gyroscopes are not uncommon [37], and it is well known that insufficient oil supply (lubricant starvation) is important to consider also in satellite attitude control systems [38,39]. However, recent research has highlighted another failure mode that is potentially significant. Rolling element bearings, critical components in these momentum wheels, might be at risk of catastrophic damage by electrostatic discharge (ESD) initiated by space weather or high voltage onboard equipment if they are lubricated by non-conductive fluids (σ < 1 pS/m) [40,41]. In tribology, it is well known that ESD can occur in bearings under certain conditions, and although various mitigation techniques exist [42,43], the risk of ESD is still an active concern that must be carefully considered on a case-specific basis in applications such as wind power [44], electric vehicles [45,46], and industrial rotating machinery [47]. ESD hazards to satellites, caused by space weather initiated dielectric charging of the spacecraft, are also well known [48]. There are mitigation techniques described thoroughly in spacecraft design handbooks and standards [49,50], but these techniques focus almost exclusively on protecting electronics or exposed external surfaces, without mentioning the risk of ESD in tribological contacts such as bearings. During validation tests, it is likely that a stationary pre-loaded bearing meets the minimum conductivity requirements designed to avoid internal dielectric charging by space weather; however, once in operation the bearings will develop a thin insulating oil film that makes them susceptible to internal dielectric charging. If an electric potential is allowed to build up until it overcomes the electric breakdown voltage of the lubricant, it may trigger an ESD that damages the sensitive bearing raceway surface. If such ESD damage is indeed responsible for some of the reliability problems encountered over the years in satellite momentum wheels [51] as suggested by Bialke [41], it could potentially be mitigated by employing electrically conductive lubricant technology, such as ionic liquids. It should be noted that even a low electric conductivity could have a large effect in comparison to a non-conductive fluid; aviation hydrocarbon fuels are commonly treated with ‘static dissipator additives’ to improve their conductivities from below 1 pS/m to a range between 50–600 pS/m [52], which is sufficient to prevent electrostatic discharge that could otherwise cause ignition during fuel transfer. In space applications, material conductivities on the order of 100 pS/m (10^12^ Ω cm) is considered sufficient to prevent spacecraft charging [50]. These values are far below the conductivities of neat ionic liquids, which are frequently in the order of mS/m [53], which indicates that even low concentrations of ILs might provide sufficient conductivity to be considered in ESD mitigation. Excessive electric conductivity may also be detrimental in other aspects, such as corrosion, and the optimal level of conductivity would certainly require tuning to the application at hand. In any case, electrically conductive lubricants have a fundamental advantage over alternative EDS mitigation approaches that are commonly applied in terrestrial applications. A conductive fluid provides a ground path that prevents buildup of electric potential, whereas an electric insulation strategy may protect the bearing without solving the fundamental issue of accumulating charge, possibly shifting the hazard of ESD along the path of least resistance to a neighboring component. Furthermore, electrically conductive fluids combined with applied electric potentials could potentially be used to improve tribological performance, as discussed in a recent review on triboelectrochemistry [54], thereby turning a loss into a gain. Clearly, space-grade lubricants—including ionic liquid concepts—play an important role in meeting the oncoming challenges in space tribology, whether in crewed space exploration, robotic planetary landers, or Earth-orbiting satellites.

### 1.3. Considerations in Lubricant Design

In order for novel lubricants to improve tribological performance in terms of reliability, efficiency, and predictability, they should be considered as an integrated part of the target tribosystem. The successful operation of such a tribosystems require a balanced interaction between all constituents of the system; the surface materials, the intermediate lubricant, as well as the surrounding atmosphere [55]. In the ideal state of machinery lubrication, the rheology of the lubricant works in concert with the elasticity of the surface material in a dynamic fluid-structure interaction that maintains a separation of the sliding surfaces by a highly pressurized (~GPa) thin (<1 µm) fluid film. This state of lubrication is ideal for high performance machinery, as friction is determined by fluid viscosity and surface damage is limited to cyclic pressure fatigue effects in the solid surfaces. This state of lubrication is known as the elasto-hydrodynamic lubrication (EHL) regime [56]. Unfortunately, EHL can only be attained with fluids that display a certain pressure–viscosity relation, and only under certain operating conditions [57]. If the fluid film thickness is reduced towards the order of the surface roughness, the surfaces boundaries will gradually come into contact in the mixed lubrication (ML) regime [58,59,60] until lubrication is dominated by surface (boundary) effects in what is known as the boundary lubrication (BL) regime. From ML to BL, tribochemical effects become increasingly significant, and chemical reactions alter the physical properties of the materials in the tribosystem. Ideally, these tribochemical reactions favorably alter the mechanical properties of the surface materials so that friction and wear are minimized, or at least controlled [61]. However, the tribochemical reactions depend on the chemistry of both the surfaces and lubricant, as well as their interaction with the surrounding atmosphere. The chemical reactions must also be tuned to the operating conditions in order to be effective, and if they are not well controlled they may worsen the tribological performance by excessive chemical reactions of the lubricant (decomposition) or surface materials (tribo-corrosion). As can be seen, the in-depth understanding of tribosystems require consideration of engineering, applied science, as well as fundamental science, across disciplines such as fluid dynamics, solid mechanics, material science, physics and chemistry [62]. The complexity of practically designing such tribosystems has forced lubricant engineers to rely heavily on empirical research in their design of lubricants. In practice, lubricants are formulated by incorporating many different additives (up to 20 wt%) into the base oil. These additives tune the lubricant performance to the specific conditions of a target application [63].

Although the empirical approach of developing base oils and additives has led to plenty of successful lubricant formulations for industrial applications on Earth, it cannot be reproduced for space-grade lubricants for several reasons. First of all, generating empirical data on space applications is cost prohibitive, and the empirical know-how used to predict terrestrial lubricant performance does not necessarily hold when significant parts of the tribosystem are altered. In fact, the space conditions may force all parts of the tribosystem to change relative to terrestrial applications, as vacuum conditions preclude the use of volatile compounds, mass constrictions promote the use of lightweight materials, and the atmosphere does not provide oxygen for the tribochemical reactions. In other words, terrestrial base oils and additives are precluded because of outgassing in a vacuum, but even if they were available they could be incompatible with the materials used, and regardless of material compatibility the lubricant tribochemistry would likely be different under oxygen-free conditions. Apart from these factors that complicate the predictability of space-grade lubricants, there are plenty of other factors to consider such as temperature, radiation, lack of maintenance, and others discussed in several excellent publications on the subject of space tribology throughout the years [64,65,66,67,68,69,70]. In any case, predicting tribological performance in space applications is challenging, and the more deviations there are from a conventional terrestrial tribosystem, the more uncertain is the prediction of tribological performance. Because of this poor predictability, spacecraft design standards recommend engineers to avoid sliding surfaces as far as possible [71], and to make use of materials and mechanisms with space mission heritage (meaning they are already qualified to work in space from previous missions) if possible [72,73,74]. Although this heritage-based approach is understandable from the point of risk mitigation, it is also further restricting the generation of empirical tribological data from space applications. As a consequence of these challenges, there are very few types of lubricant qualified for use in space applications in comparison with the breadth of options for terrestrial applications.

### 1.4. Current State of Space-Grade Lubricants

Multiply alkylated cyclopentanes (MAC) [75] and perfluoroalkyl polyethers (PFPE) [76] are two liquid lubricants with significant space heritage. These fluids display low enough vapor pressure to ensure they meet the strict performance requirements related to outgassing that is required to certify liquid lubricants for use in space applications [77]. Although these fluids display high thermal stability and chemical inertness under static conditions, both MAC and PFPE are prone to tribochemically induced decomposition under BL conditions. PFPE is known to form iron fluorides under BL which auto-catalytically decomposes the fluid [78], while MAC is catalytically decomposed by nascent iron surfaces generated by wear in oxygen-free environments [79]. Lubricant additives, therefore, serve a dual purpose as they protect the solid surfaces by generating a boundary film, and simultaneously protect the base fluid from decomposing by exposure to nascent metal [80,81]. Unfortunately, few additives are available for MAC or PFPE, since any effective additive needs to (i) overcome the problem of dissolving polar additive in non-polar base oil, (ii) have a low enough vapor pressure to avoid excessive evaporation under vacuum, (iii) be effective in an oxygen-free environment. Additive miscibility is significantly worse in PFPE than MAC, mostly because PFPE has a chemical structure very different from that of conventional hydrocarbon fluids, which existing additives were designed for. It is possible to disperse solid lubricants in PFPE [82], but conventional additives are not miscible. There are some additive options available for use with MAC, including conventional additives [83,84,85,86], solid dispersions [86,87,88,89], and ionic liquids dissolved in MAC [90,91], however, the combination of miscibility, low volatility, and tribological performance remains problematic. To improve additive solvency in conventional synthetic hydrocarbons like PAO (polyalphaolefin), it is common to blend the base fluid with a polar base fluid such as an ester [92], but this option is not available to MAC as it would negatively affect the vapor pressure. Fully oil-miscible ionic liquids are available [22], however, a high degree of alkylation correlates with increased vapor pressure, and such ionic liquids display pronounced outgassing in vacuum and decreased thermal stability [18,19].

To summarize the state of liquid lubricants in space tribology, there is no silver bullet found, however, more options are becoming available for space-grade lubricant formulators, although the state is still far from terrestrial applications. Many conflicting features need to be balanced against each other. For example, ILs that mimic hydrocarbons display improved compatibility with existing tribological systems including base oils, but if the IL is too closely mimicking hydrocarbons, it may lose the benefits of ILs such as low vapor pressure and high thermal stability. Similarly, boundary film formation requires reactivity between the additive and the surface material, but too much reactivity may cause tribo-corrosive wear. The lubricant should also display some electric conductivity in order to avoid static charge buildup and electric discharge damage, however, excessive conductivity may promote electrochemical reactions that degrade the tribological system. The perfect lubricant formulation depends on the exact application at hand, and for this reason, tribology engineers need to have a range of lubricants and additives at hand, in order to tailor the lubricant system to the specific problem.

### 1.5. Contribution of This Work

This work builds upon our previous research, where the principle of molecular design of lubricants [93,94] was employed to design an ionic liquid specifically for use in lubrication [95]. The resulting IL, composed of tetraalkylphosphonium cation and a trimethylsilylalkylsulfonate anion, was designated P-SiSO. This IL attempts a balanced approach of reactivity to tribomaterials, as well as a balanced oleophilic index that mimics the hydrocarbon structure while retaining important bulk properties, such as low vapor pressure. In our previous work, we found excellent tribological performance attributed to the formation of a silicate boundary film [96]. P-SiSO displayed apparent miscibility at 1% wt in squalane and excellent tribological effectiveness when evaluated as additive in air under rolling-sliding conditions [97], as well as under component scale testing of geared actuators in our ongoing work [98].

In this paper, we specifically investigate the suitability of employing P-SiSO as an additive in MAC to improve performance in the boundary lubrication regime, while simultaneously providing a well-balanced degree of electric conductivity of the lubricant. The experimental investigations in this paper focus on outgassing, conductivity, and tribological performance. The tribological experiments investigate the friction and wear sensitivity to atmospheric conditions by testing under air, dry N_2_, or dry CO_2_, with validation in high vacuum. 3D-surface analysis is conducted by scanning white light interferometry, followed by scanning electron microscopy with energy-dispersive X-ray spectroscopy (SEM-EDS) for characterization of boundary films.

## 2. Materials and Methods

### 2.1. Lubricant Samples

An experimental lubricant, designated ‘MAC &P-SiSO’, was prepared by dissolving 0.4 wt% of the ionic liquid P-SiSO (trihexyltetradecylphosphonium 2-(trimethylsilyl)ethane-1-sulfonate) [95], in the base fluid MAC, a multiply alkylated cyclopentane (Synthetic Oil 2001a, supplied by Nye Lubricants, Inc. (Fairhaven, MA, USA)). The solution was tempered at 70 °C for 30 min, followed by sonication for 30 min. The homogenous solution of ‘MAC &P-SiSO’ was confirmed by subjecting samples to centrifuging at >15,000 G acceleration, followed by sampling of top and bottom phase for characterization by Fourier-transform infrared spectroscopy (FTIR). The higher viscosity of P-SiSO compared to MAC was assumed to have no significant influence on the bulk viscosity of the ‘MAC &P-SiSO’ mixture at the concentrations used in this work. This assumption was verified by use of Equation (1) which is used to calculate the viscosity of the mixture, $\nu_{mix}$ [22]. Viscosity data for MAC and P-SiSO at 40 °C is denoted as $\nu_{MAC}$ and $\nu_{IL}$ respectively, and their relative concentrations are denoted $X_{MAC}$ and $X_{IL}$.
(1)$$\nu_{mix}=\exp\left(\exp\left(X_{MAC}\log\left(\log\left(\nu_{MAC}+0.8\right)\right)\;+\;X_{IL}\times \log\left(\log\left(\nu_{IL}+0.8\right)\right)\right)\right)\;-\;0.8$$

With the numeric values provided in the Appendix A we calculate a negligible viscosity increase of <0.5% in ‘MAC &P-SiSO’ compared to neat MAC at the temperatures used in the tribology experiments.

For the tribological experiments, we employed neat MAC, neat P-SiSO, as well as neat PFPE, a perfluoropolyether lubricant (Fomblin Z25, a supplied by Solway S.A. (Brussels, Belgium)) as reference lubricants.

### 2.2. Conductivitiy Experiment

‘MAC &P-SiSO’ (0.4 wt% P-SiSO in MAC) was freshly prepared following the procedure described in Section 2.1 Lubricant Samples to obtain a homogenous solution, which was used to determine its conductivity value. The dependence of conductivity, together with relative permittivity on temperature was determined at 25 °C, 50 °C, 75 °C and 100 °C temperature steps. Permittivity describes the material ability to transfer electrical field and is expressed in Farad per meter. However, in our set-up we measured relative permittivity as dimensionless value as it compares the measured value to vacuum permittivity. The measurements were carried out by an in-house developed alternating current (AC) conductivity measurement device consisting of two gold coated electrodes separated by 1 mm gap of 2100 mm^2^ surface area. An alternating square wave voltage of ±2.5 V was applied with frequency of 0.5 Hz during the conductivity measurements. In the case of relative permittivity measurements, a sin wave voltage of ±2.5 V was applied with a frequency of 2 kHz. The measurement device was calibrated using air, toluene, n-heptane and diethyl ether.

The purpose of the analysis was to evaluate the feasibility of achieving a relevant level of electric conductivity by the incorporation of a low concentration of ionic liquid in a heritage space lubricant. Conductivity levels in the order of 100 pS/m were considered relevant in this work, based on engineering standards and guidelines applicable to similar systems and applications [50,52].

The electric conductivity, σ was expected to vary with temperature, *T*, as the charge-conducting ions must overcome the viscous resistance that is highly temperature dependent. This effect is well known for low concentrations of conductive additives employed in hydrocarbon solvents. The relation of conductivity, σ, to temperature can be described by a semi-log relationship that is derived by measuring conductivity for at least two different temperatures (*T1*, *T2*) to retrieve a temperature-conductivity coefficient, *n* [°C^−1^] [99], shown in Equations (2) and (3).
(2)$$\log_{10}\sigma_{T1}=n\left(T1-T2\right)\;+\;\log_{10}\sigma_{T2}$$
(3)$$n=\frac{\log_{10}\left(\frac{\sigma_{T1}}{\sigma_{T2}}\right)}{T1-T2}$$

### 2.3. Outgassing Experiment

The outgassing of the lubricants was assessed in accordance with the European Cooperation for Space Standardization (ECSS) space product assurance standard “thermal vacuum outgassing test for the screening of space materials”, ECSS-Q-70-2C [77]. This test quantifies total mass loss (TML), relative mass loss (RML), and collected volatile condensable material (CVCM) as performance indicators of liquid sample outgassing under thermal vacuum conditions. TML and RML are directly related to the expected useful life of the lubricant in a space application, whereas CVCM is related to risk of contamination by the outgassed compound. The general acceptance limits are stated as TML and RML <1.0 %, and CVCM <0.10%. Depending on the exact application, these limits may be more or less stringent, however these values serve as a useful indicator of the suitability of applying these fluids as lubricants in space applications. This European standard is equivalent to the U.S. standard ASTM-E595 [100].

Three lubricants were evaluated; neat MAC (as reference), neat P-SiSO, and the experimental lubricant ‘MAC &P-SiSO’ which was prepared to contain 0.4 wt% P-SiSO dissolved in MAC. A total mass of 0.55 g for each sample was evaluated, divided into three specimen cups with approximately 180 mg of the specimen in each cup. Three empty cups was also prepared as references, and collector plates were used to measure the collected volatile condensed material. The outgassing experiment was conducted over 24 h, with 24 h conditioning phases before and after the test. During the conditioning phase before the test, the samples were maintained for 24 h in high humidity (55 ± 10% RH) at room temperature (22 ± 3 °C) and ambient pressure. After the samples were conditioned, they were weighed on a precision balance before they were mounted in the vacuum chamber. In the subsequent outgassing experiment, the test chamber was evacuated to a pressure below 10^−3^ Pa, while heating the samples to 125 °C. These conditions were reached within the first hour, and maintained over the remaining time of the 24 h experiment. The samples and collector plates were then removed from the vacuum chamber and weighed again. The samples were finally conditioned for 24 h in high humidity (55 ± 10% RH) at room temperature (22 ± 3 °C) and ambient pressure, before being weighed again. Further details of the method used can be found in the standard ECSS-Q-70-2C [77].

### 2.4. Tribology Experiments

The influence on friction and wear by the lubricants was evaluated by two types of sliding point-contact tribometers; (i) The MVT-2 multi-functional vacuum tester (RTEC instruments, San Jose, CA, USA) was used in unidirectional sliding ball-on-disc configuration under high vacuum (HVAC) conditions to investigate the feasibility of lubrication under vacuum conditions with the investigated lubricants. Secondly, (ii) the Optimol SRV-3 reciprocating ball-on-flat tribometer (Optimol Instruments Prüftechnik GmbH, München, Germany) was modified with gas inlet to provide options for control of atmospheric conditions (Air, N_2_, or CO_2_), and was employed to study the performance under a wide range of conditions, and to prepare samples suitable for boundary film analysis.

The tribological performance was quantified based on the friction force continuously measured by the tribometers, and by a normalized wear index that compares the diameter of the wear scar on the ball (WSD) with the nominal Hertz contact diameter (HzD), as explained previously [96,101] and reiterated in the Appendix A. Surface analysis was performed on selected samples by light optical microscopy (LOM), 3D profilometry (3DP), and SEM-EDS.

In all experiments, the test duration was 30 min, and steel specimen of AISI 52100 were used for both ball and disc. The test conditions were defined by a combination of applied load, F, the resulting maximum contact pressure, P_max_, and the chosen atmospheric conditions, ATM, and will be reported in the following format: {F/P_max_/ATM}. Hertzian contact pressures (P_max_) and diameters (HzD) are calculated by Hertz theory, taking into account the geometrical and mechanical properties of the steel samples used [102]. Table 1 summarizes important parameters that differed for the experiments with the two different tribometers. We relied on the film parameter, Λ, to ensure that the lubricant performance was evaluated in the boundary lubrication (BL) regime (Λ<1) [57]. The film parameter was retrieved in two steps. First, the lubricant film thickness, $h_{min},$ was calculated by the Dowson–Hamrock equation that takes into account the effective contact radius ($R_{x}$) and ellipticity (k), as well as dimensionless parameters related to speed (U), materials (G), and load (W), as shown in Equation (4). Secondly, the film thickness was compared to the root mean square surface roughness, Sq, as shown in Equation (5). The numeric values for the different cases are found in the Appendix A.
(4)$$h_{\min}=R_{x}\;\cdot \;3.63\;\cdot \;U^{0.68}G^{0.49}W^{-0.073}\left(1-e^{-0.68k}\right)$$
(5)$$\Lambda =\frac{h_{\min}}{\mathrm{Sq}}$$

Using this approach, we compensate for the different composite surface roughness (Sq) of the specimen types used for the two tribometers. The smoother surfaces used in the MVT-2 experiments was compensated for by adjusting the conditions to lower sliding speed and slightly elevated temperature in relation to the SRV-3, as seen in Table 1. Further information of the experiments are given separately for each tribometer in the following sections.

#### 2.4.1. MVT-2 Tribotest in Vacuum

The MVT-2 vacuum tribometer was used to evaluate baseline lubricant performance in a vacuum environment. A 6.35 mm steel ball and disc specimen of AISI 52100 steel was used. The standard specimen used in the MVT-2 were of significantly lower surface roughness compared to the standard specimen used in the SRV-3 experiments, with the MVT-2 specimen having a composite roughness (Sq) of only 9 nm compared to 61 nm for the SRV-3 specimen. In order to avoid excessive hydrodynamic effects by the smooth surfaces, and to facilitate the comparison with SRV-3 experiments in the boundary lubrication regime, the test conditions were chosen so that the minimum lubricant film thickness was ensured to be lower than the composite surface roughness as described in the previous section. With the conditions stated in Table 1 the minimum film thickness was calculated to be 8 nm based on the Dowson–Hamrock equations, thus ensuring that the test was initiated in the boundary lubrication regime (Λ<1). The numerical data for the calculation of the film parameter (Λ) is provided in the Appendix A.

The test procedure for the MVT-2 experiments was executed as follows: (I) A sufficient amount of lubricant (0.3 mL) was applied onto the test disc to ensure that the whole track around the disc would have similar amount of lubricant supplied. (II) High-vacuum (HVAC) conditions were set by evacuating the closed test chamber until the indicated pressure was below 5 × 10^−3^ Pa. (III) Thermal stabilization and lubricant distribution was ensured before starting the experiment. The ball was commanded into light contact (1 N) with the disc at a radius of 8 mm while the disc was rotating (10 rpm), in order to distribute the lubricant along the track. The thermostat was set to 40 °C and the temperature stabilized within 5 min. (IV) The test was initiated by increasing the speed to 60 rpm (stabilized within 1 s), which corresponded to 0.05 m/s, while increasing the load to 40 N (stabilized within 5 s), which corresponded to 2.1 GPa. The test duration was 30 min. (V) after the test finished, the ball and disc was washed ultrasonically in heptane and ethanol and characterized by surface analysis.

#### 2.4.2. SRV-3 Tribotest in Controlled Atmosphere

The SRV-3 was used for screening of lubricant performance characteristics over various atmospheric and contact pressure conditions, and for generating boundary film samples suitable to analysis by EDS. Three types of atmospheric conditions were used in this setup, denoted as AIR, N_2_, and CO_2_. AIR is defined as ambient air, having a relative humidity between 15–50%. N_2_ conditions is defined as continuous flooding of the test chamber with dry gaseous nitrogen so that the measured levels of oxygen and relative humidity were maintained below 5%. Finally, the CO_2_ condition is defined as continuous flooding of the test chamber with dry CO_2_ gas that maintained the measured concentration of CO_2_ above 90%, while relative humidity remained below 5%. The load conditions were set to either 150 N load, or 300 N load, which resulted in maximum contact pressures (P_max_) of 2.4 and 3.0 GPa, respectively. All the experiments using SRV-3 was conducted at ambient temperature (25 ± 2 °C).

The experimental procedure was executed as follows: (I) Before each test, we applied a small amount (30 µL) of the test lubricant into the sliding interface. (II) The atmospheric conditions were set by flooding the test chamber with AIR, N_2_, or CO_2_, as defined previously. (III) The 10 mm steel ball was pressed against the flat steel disc at the selected applied load (150 or 300 N), and after ensuring sample alignment the test was initiated. (IV) During the 30-min test duration, the ball was subjected to reciprocating motion at 50 Hz with a stroke length of 1 mm while the selected applied load (150 or 300 N) was maintained. (V) after the test finished, the ball and flat was washed ultrasonically in heptane and ethanol and characterized by surface analysis. Each test was repeated twice.

The procedure is based on ASTM D6425 and the sample specimen used conforms to the specification therein [103]. The composite roughness of the surfaces was calculated to 61 nm based on the specified sample roughness, and the Λ<0.52, as listed in Table 1. Furthermore, the conditions used ensured that the relation between Hertz contact diameter (HzD) and stroke length was sufficient to avoid fretting motion.

### 2.5. Analysis of Worn Surfaces

A Zygo NewView 7300 optical 3D profilometer (3DP) (Zygo Corporation, Middlefield, CT, USA) was employed for surface analysis by scanning white light interferometry. The surfaces were measured using 10× and 50× Mirau objectives, yielding maximum resolutions <0.5 µm spatially and <0.1 nm vertically. The surface profilometry data was analyzed using MetroPro 9.1.6 (Zygo Corporation, Middlefield, CT, USA), and MountainsMap Premium 7.4 (Digital Surf, Besançon, France). A Dino-Lite AM7915MZTL 5MP digital light optical microscope (LOM) (Dino-Lite Europe, 1321 NN, Almere, The Netherlands) was also employed for surface analysis in the visible spectrum of balls and discs.

The worn regions on the discs were further analyzed by field-emission scanning electron microscopy (SEM) in low voltage high-contrast detector mode (vCD) at 3 kV, using a Magellan 400 FEG-SEM (FEI Company, Eindhoven, The Netherlands). Energy-dispersive X-ray spectroscopy (EDS) was performed using an X-Max 80 mm2 X-ray detector (Oxford Instruments, Abingdon, UK) operated at 5 kV, which is enough to detect elements of interest such as P, Si, S, O, C, and Fe, while maintaining a minimal depth of penetration. The EDS-spectra of boundary films developed in the tribological contacts were compared to EDS-spectra of unused reference test specimen (AISI 52100 steel). As seen in Figure 1, the microstructure of the reference steel contains features that are rich in the elements that are also present in P-SiSO, such as Si, O, and C. These features can be clearly distinguished from the boundary films formed by P-SiSO by their respective visual appearances under the scanning electron microscope (SEM), especially when using the vCD solid state detector. Nonetheless, we prepared two reference spectra representative of the AISI 52100 bulk steel substrate (Ref) and silicon-carbon containing surface features (SiC) to assist in the analysis of boundary films. The spectra Ref and SiC shown in Figure 1b are the results of averaging the spectra at Ref-1–3 and SiC-1–2 respectively. Elemental analysis of the boundary films are performed by dividing the boundary film spectra with the reference spectra, so that the relative abundance of elements in the boundary film can be retrieved.

## 3. Results and Discussion

### 3.1. Outgassing Analysis

The results of the ECSS-Q-70-2C outgassing experiment are shown in Figure 2. As expected, neat MAC performed well, and all outgassing quality parameters (TML, RML, CVCM) was below the acceptance limits for materials to be used in spacecraft [77,100]. The results of ‘MAC &P-SiSO’ clearly show that the addition of 0.4 wt% P-SiSO to MAC did not adversely affect the outgassing performance, hence ‘MAC &P-SiSO’ also meets the outgassing requirements for space grade materials. For neat P-SiSO, the total mass loss (TML) and collected volatile condensed material (CVCM) are above the standard acceptance level, while the relative mass loss (RML) is below the acceptance limit. The difference between TML and RML is commonly referred to as water vapor regained (WVR) in ECSS-Q-70-02C. This result indicate that the outgassed species from neat P-SiSO is mainly water vapor that was absorbed during the conditioning step. Although highly alkylated ILs (including tetraalkylphosphonium) are considered hydrophobic, they can display significant water absorption in a humid atmosphere [104]. Regarding the relative mass loss of P-SiSO, it indicates that there is measurable outgassing, albeit at a rate that is acceptable according to the ECSS-Q-70-02C standard. It is reasonable to assume that humidity is detrimental to the vapor pressure of P-SiSO, as hydrolysis of P-SiSO can produce a neutral ion pair with high boiling point via the reaction in Equation (6).
(6)$$\left[R_{4}P\right]\left[{\mathrm{TMSC}}_{2}H_{4}{\mathrm{SO}}_{3}\right]\;+\;{\;H}_{2}O⇌\;\left[R_{4}P\right]\left[\mathrm{OH}\right]\;+\;{\;\mathrm{TMSC}}_{2}H_{4}{\mathrm{SO}}_{3}H$$

Formation of neutral ion pairs are known to increase the vapor pressure of ionic liquids [105], and should be avoided if possible. Incorporating anti-oxidants is a feasible strategy of mitigation, and in our previous work we successfully employed N-Phenyl-1-naphthylamine as an additive in P-SiSO [96].

The maximum allowable concentration of P-SiSO in MAC can be estimated by interpolating the outgassing performance of neat P-SiSO and neat MAC. Assuming there are no antagonistic effects, and sufficient solvency, it would be possibly to use a concentration of approximately 10 wt% of P-SiSO in MAC without compromising the outgassing performance. Our preliminary analysis indicates that the maximum allowable concentration of P-SiSO in MAC would be limited by solvency before exceeding the acceptable limits of outgassing performance.

### 3.2. Conductivity Analysis

The conductivity analysis of ‘MAC &P-SiSO’, containing 0.4 wt% P-SiSO in MAC, is summarized in Figure 3. We measured electric conductivity over a range of temperature from 25 °C to 100 °C. The conductivity measurements were made sequentially, starting at 25 °C and repeated at temperature increments of 25 °C until the 100 °C level was reached, before allowing the liquid to cool down to 25 °C to measure the conductivity again, as displayed in Figure 3a. A strong temperature-dependency was expected—mainly due to the influence of viscosity—and can be estimated according to ASTM D2624 as described in Equations (2) and (3). However, apart from the temperature influence, the measured conductivity also displayed an obvious time-dependency at higher temperatures. Starting at 75 °C, we observed that conductivity was increasing over time before stabilizing within a few minutes, as indicated by the arrow in Figure 3a. Based on this observation of time-dependency, the conductivity was recorded continuously at the 100 °C temperature level as shown in Figure 3b. During this measurement, the conductivity continued to increase for ~90 min before finally reaching a steady state at 382 pS/m. Considering the fact that the temperature had stabilized within 5 min, the results shown in Figure 3b clearly show that temperature effects on viscosity is not sufficient to explain the observed time-dependent increase in conductivity. The final measurement of the sequence displayed in Figure 3a was the conductivity re-measured at 25 °C after allowing the liquid to cool down from 100 °C. This measurement provides further information about factors that influence the conductivity of this system. Clearly, the re-measured conductivity at 25 °C is significantly higher than what was measured in the initial state at the same temperature. The observed temperature-hysteresis of conductivity produced an increase in conductivity from 22 to 96 pS/m, an increase of a factor of almost 4.4 between the initial measurement at 25 °C and the final (steady) measurement at 25 °C.

The investigated liquid, ‘MAC &P-SiSO’, can be modeled as a simple ionic electrolyte system with a low concentration of large ions in an apolar solvent. Considering classical theory of ion-solvent dynamics, ionic conductivity of such an electrolyte depends mainly on the ability of the charge-conducting ions to overcome the viscous resistance of the solvent [106]. The increase in conductivity should therefore be an effect of (i) a decrease in viscosity, (ii) an increase in the ionic mobility, or (iii) an increased number of charge carriers, or a combination thereof. Theoretically, decomposition of the base fluid or ionic liquid could influence these factors. Therefore, post-experiment FTIR analysis was performed and compared to pre-experiment FTIR analysis and, as shown in Figure 3c, we did not observe any sign of degradation, as the ‘Before’ and ‘After’ FTIR-results could not be distinguished from each other. This indicates that it is unlikely that any permanent degradation—sufficient to explain the 4.4 times increase in conductivity—occurred at elevated temperatures, especially considering the high chemical stability of the investigated compounds. Based on this analysis, viscosity effects cannot explain the increase in conductivity seen in the initial and final measurement at 25 °C, or the time-dependency observed at 100 °C, and thus option (i) can be ruled out as an explanation of the increased conductivity. The remaining options are (ii) increased ionic mobility, or (iii) an increased density of charge carriers. Option (ii), increased ionic mobility, is also theoretically possible by decomposition if it causes a reduction of the effective ion radius. The effect on ion radius on ion mobility can be estimated by the Stokes–Einstein equation as explained in [53]. However, also in this case the decomposition would have to be significant to explain the 4.4 times increase in conductivity, and such a decomposition should be detectable by FTIR. Consequentially, option (ii) is also unlikely as an explanation of the increased conductivity. The remaining alternative to explain the observation of increasing conductivity at constant temperature is option (iii), an increased density of effective charge carriers. The number of charge carriers depend on the amount of charge-carrying mobile species, and the presence of ionic aggregates or ion pairs form neutral species which cannot be considered charge carriers [107]. Furthermore, ionic mobility can be inhibited by adsorption, which would also decrease the effective number of charge carriers. This mechanism is also in line with reports of time-dependent conductivity for lubricating oils with conductivities in the range of 1–500 pS/m, which was attributed to ionic mobility factors [108]. Under an alternating electric field—as was employed in our experiment—it is reasonable to assume that ions of opposite polarity will strive to reposition into an energetically favorable ordered distribution, as shown schematically in Figure 3d. It is also reasonable to assume that the rate of this process is increased at higher temperature, due to reduced viscosity. This mechanism would explain the conductivity-temperature hysteresis that was observed upon cooling the fluid down from 100 °C to 25 °C.

To estimate the conductivity in a practical application such as a ball bearing at operating speeds is complicated by the factors discussed. The effect on conductivity by factors such as lubricant confinement, high pressure and shearing motion are not obvious. However, it is clear that the magnitude of the conductivity is in the order of 100 pS/m, which was the criteria for being considered relevant in minimizing the risk of electrostatic discharge damage in a space application.

### 3.3. Friction and Wear Analysis

#### 3.3.1. MVT-2 Tribotest in Vacuum

Results from the MVT-2 vacuum tribometer experiments are shown in Figure 4. These experiments evaluated neat MAC and ‘MAC &P-SiSO’ under high vacuum (HVAC) conditions (<10^−3^ Pa), at 40 °C and 40 N load (2.1 GPa) under unidirectional sliding conditions. As explained in Section 2.4.2 under Materials and Methods, these experiments were conducted with the highly polished standard MVT-2 samples, having a composite roughness below 10 nm. The conditions were adapted to minimize hydrodynamic effects and focus on the boundary lubrication performance. The friction traces of neat MAC and ‘MAC &P-SiSO’ over the 30-min experiment duration are displayed in Figure 4a. Figure 4b provides the 3DP topographical data of the ball wear scars, together with numeric wear indices (W_Hz_) and wear volumes (W_v_). The ball form having a radius of 3.175 mm has been removed from the 3DP ball topography data. Figure 4c–e display detailed views of the running-in (c) period, as well as two indications of partial seizures observed at the time (d) 1090 s and (e) 1540 s into the experiment with neat MAC.

As seen in Figure 4a, and magnified in Figure 4c, the neat MAC displays a pronounced running-in behavior with a rapidly rising friction in the initial 10 s. A significantly increased friction coefficient is maintained during the initial 4 min (240 s) before stabilizing around the initial value. It is deemed likely that higher friction during the running-in phase was accompanied by significant wear. In contrast, ‘MAC &P-SiSO’ display a relatively steady running-in process, with a very slight increase in friction from 30–60 s into the experiment. However, also with ‘MAC &P-SiSO’ there was a running-in process over the initial ~500 s. In the case of ‘MAC &P-SiSO’, the friction coefficient remains low, but the running-in process is accompanied by slight fluctuations in friction before the friction stabilizes. The 3DP data reported in Figure 4b display wear indices (W_Hz_) of 1.21 for neat MAC, and 1.05 for ‘MAC &P-SiSO’. The removed volume of material (W_v_) was twice as large in the sample lubricated by neat MAC as compared to ‘MAC &P-SiSO’. It is likely that a significant part of the wear seen with MAC was produced during the running-in phase, as indicated by the significantly higher friction. After such a period of high wear, the load-bearing contact area increases, which in turn lowers the actual contact pressure throughout the remaining test.

Regarding the indication of partial seizures in neat MAC highlighted in Figure 4d–e, this behavior is not surprising considering again that neat MAC does not contain any triboimproving additive. As seen from the 3DP data, the wear depth is >0.4 µm, indicating that initial oxide layers are likely worn through at this point. In oxygen-depleted conditions, nascent metal exposed by wear will remain reactive for longer than it would in air, leading to an increased risk of severe adhesion. In space tribology, this is often referred to as cold welding, and partial seizures are to be expected [109]. Under the conditions of this experiment, wear on the rotating disc was barely measureable as it was distributed over the entire arc length of the wear track. In the case of a longer test duration, it is possible that disc oxide layers would also be removed, with ensuing risk of gross seizure by cold welding. In contrast to the partial seizures observed in the case of neat MAC, there were no such observations in the case of ‘MAC &P-SiSO’.

#### 3.3.2. SRV-3 Tribotest in Controlled Atmosphere

Figure 5 summarizes the SRV-3 tribological results in terms of friction coefficient (µ) and wear index (W_Hz_), with the wear index being defined as the wear scar diameter divided by the Hertz contact diameter (W_Hz_ = WSD/HzD). Figure 5a display the friction traces recorded for neat MAC, neat PFPE, ‘MAC &P-SiSO’, and neat P-SiSO at conditions of {150 N/2.4 GPa/N_2_}. Every data point is the average friction coefficient over 1 s, and every test was repeated at least twice. Friction traces for all six conditions evaluated (loads 150 or 300 N, atmospheric conditions AIR/N_2_/CO_2_) are available in the Appendix A. Figure 5b include pairs of ball and disc wear scars for neat MAC, ‘MAC &P-SiSO’, and neat P-SiSO, as seen with an optical 3D-profilometer (3DP). Color-coded topography data is overlaid on 3DP intensity charts to provide representative images of the morphology and severity of wear encountered on both ball (upper part of image) and disc (lower part). The 5 mm ball radius has been removed from the 3DP ball topography data. Further images can be found in the Appendix A. Figure 5c–h provide the average friction coefficient (µ_avg_) over the full test duration, plotted against the average wear index (W_Hz_). The dashed line represent the theoretical minimum wear index (W_Hz_ ≥ 1), and seizure is defined as sustained friction force above 100 N (µ above ~0.33 at 300 N).

The results show that the addition of 0.4 wt% P-SiSO in MAC has a significant improvement on both friction and wear performance in all the conditions evaluated. As seen in Figure 5a, neat MAC display poor friction characteristics under these harsh boundary lubricated conditions, with a severe increase in friction during the running-in period followed by high and unsteady friction throughout the test. Under the same conditions {150 N/2.4 GPa/N_2_}, neat P-SiSO performs well, with very smooth friction characteristics and low wear, similar to the results seen in previous studies of neat P-SiSO under air conditions [95,96]. ‘MAC &P-SiSO’, containing 0.4 wt% P-SiSO in MAC, prevents the abrupt friction increase seen with neat MAC during the running-in phase, and reduces friction and wear significantly during the test. In fact, as seen in Figure 5c–h, ‘MAC &P-SiSO’ reduces friction and wear significantly under all the evaluated conditions in comparison to neat MAC as well as the reference lubricant PFPE.

The wear scars displayed in Figure 5b indicate abrasive wear along the direction of sliding in the case of neat MAC, which is likely initiated by adhesive damage already in the running-in phase. In contrast, the wear scar topography produced with ‘MAC &P-SiSO’ resembles that of neat P-SiSO, with a very smooth surface remaining at the end of the test. As seen in the 3DP intensity charts, the dark color indicating boundary film is slightly more evident in the case of ‘MAC &P-SiSO’ in comparison with neat P-SiSO. Comparing Figure 5c–d, the effects of AIR or N_2_ conditions are minor. All lubricants display comparable performance in AIR conditions as in N_2_, however, the performance of neat MAC was significantly improved in the CO_2_ conditions as seen in Figure 5e, which indicates effective boundary film formation by neat MAC in CO_2_ atmosphere. The results under higher load (300 N) seen in Figure 5f–h, show that the performance increase by CO_2_ was not enough to prevent immediate seizure, which was observed for neat MAC in all atmospheric conditions under high load (300 N). Nonetheless, the effect on tribological performance with MAC under CO_2_ might be of interest in the search for triboimproving concepts for MAC.

The results at higher loads shown in Figure 5f–h display that the addition of 0.4 wt% P-SiSO in MAC also prevents the immediate seizures observed with neat MAC, although seizures and partial seizures was observed under N_2_ and CO_2_ atmosphere respectively with ‘MAC &P-SiSO’. Neat P-SiSO performed well in all conditions evaluated, maintaining a low friction coefficient as well as a low wear index under all evaluated conditions. Partial seizures were observed also with PFPE under AIR and CO_2_ conditions, whereas the friction was more stable under N_2_ conditions. PFPE under boundary lubrication is known to react with exposed (nascent) Fe, forming iron fluorides that can act as extreme pressure agents that reduce friction at the expense of high wear. Under AIR and CO_2_ conditions, it is likely that the formation of iron fluorides is competing with formation of iron oxides as nascent Fe is exposed. In any case, FeF_3_ is problematic as a boundary film compound as it is acidic and can catalytically decomposes the PFPE into low molecular weight volatiles [78], as well as corrode the metal in the presence of humidity [110].

It is clear that atmospheric conditions also influence the performance of P-SiSO. Neat P-SiSO displayed slightly improved tribological performance under N_2_ compared to AIR, whereas ‘MAC &P-SiSO’ displayed an opposite trend with better performance in AIR compared to in N_2_ conditions. The best overall lubricating performance was observed with neat P-SiSO under N_2_ conditions.

### 3.4. Boundary Film Analysis of SRV-3 Samples

#### 3.4.1. Surface Analysis by Complementary Techniques

We employed a set of complementary techniques (LOM, 3DP, SEM-EDS) in order to gain insight into the boundary films formed by P-SiSO in the neat form or as a lubricant additive in MAC at 0.4 wt% concentration. Worn samples from SRV-3 experiments were analyzed. The main focus of the study was ‘MAC &P-SiSO’, while neat P-SiSO and neat MAC was analyzed as references. LOM was used to record the visual appearance and spatial extent of the boundary film coverage. The 3DP technique based on scanning white light interferometry provides topography information that quantify the amount of wear, and 3DP was also employed to provide information about the extent of boundary film coverage by analyzing the intensity of the reflected light over the analyzed area. Finally, SEM provided details of the surface morphology that were necessary to understand mechanisms of wear, while EDS provided elemental chemical information that could be used to infer the boundary film composition. The EDS signal is normalized against the average background radiation measured between 1.0–1.5 keV in the energy spectra.

The analysis in Figure 6, Figure 7 and Figure 8 focuses on SRV-3 samples subjected to conditions {150 N/2.4 GPa/AIR}, however, a similar analysis was made for all conditions and further results can be found in the Appendix A. All analyzed samples shown were washed ultrasonically in two steps before the analysis; five minutes immersed in heptane followed by five minutes immersed in ethanol.

Figure 6 include surface analysis of neat MAC by LOM, 3DP, and SEM-EDS. When studying the worn disc surface lubricated by neat MAC under LOM, as seen in Figure 6a, there is no indication of boundary film present. The surface appears brighter than the surrounding unworn material, which is indicative of nascent steel exposed by severe wear. The ball wear scar image seen to the left in Figure 6a was produced by 3DP in intensity mode, and the dark appearance of the wear scar is in this case an effect of reduced reflection caused by the significant roughness of the surface. Figure 6b shows a detailed view of the ball topography measured by 3DP. The ball curvature has been subtracted from the data in order to discern the depth of wear on the ball (~5 µm). It is clear that most of the wear is located at the center of the ball, where the contact pressure is highest. This result is in line with the well-established Archard wear model, which predicts that wear is proportional to load [111] and, therefore, wear depth should correspond to the pressure distribution. The disc topography measured by 3DP is seen in Figure 6c–d at low and high magnification. It can be clearly seen that lubrication by neat MAC under the conditions {150 N/2.4 GPa/AIR} produces a rough surface that appears severely abraded. The detailed view in Figure 6d show that there is a significant amount of adhered material on the disc surface, leading to a high area roughness (Sq = 0.317 µm). Figure 6e show the same region by SEM, with five locations (approximately 1 × 1 µm) selected for EDS analysis. The EDS-analyzed regions were selected to search for evidence of chemical reactions, although in the case of neat MAC in AIR, no boundary film was expected. As seen in Figure 6f, the detected energy peaks (normalized counts) are associated with carbon, oxygen, and iron, and there is no significant difference compared to the unworn reference shown in Figure 1. Also, there is no significant difference between the regions marked ‘Film’ or ‘Blank’ (plotted signals are shifted vertically by 4 units to improve visibility).

This analysis shows that the neat reference lubricant MAC behaves as expected under these conditions, with high friction and wear. No significant boundary film was produced to protect the surfaces from severe adhesion.

The boundary film formed by the lubricant consisting of 0.4 wt% P-SiSO in MAC, ‘MAC &P-SiSO’, is analyzed in Figure 7. When studying the worn surfaces under LOM, they appear covered in blue, purple and black patches, indicative of boundary film. The detailed 3DP-view of the ball wear scar topography shown in Figure 7b indicate that the central load carrying region of the ball has been well protected, in contrast to the surface lubricated with neat MAC seen in Figure 6b. Clearly, the wear mechanism when lubricated with ‘MAC &P-SiSO’ no longer follows the simple relation of wear rate proportional to load as described by the Archard wear model. From Figure 7c–d, it is clear that the disc surface has experienced little wear, and the roughness was actually slightly reduced within the wear scar (Sq = 0.102 µm) compared to the unworn surface outside the wear scar (Sq = 0.116 µm). The SEM-EDS analysis in Figure 7e–f indicates a boundary film composed mainly of silicate, which is in line with results seen in previous work [96,97].

The analysis of the boundary film formed by neat P-SiSO is shown in Figure 8. The visual appearance of the disc wear scar as seen by LOM in Figure 8a indicates the presence of boundary film. In comparison to ‘MAC &P-SiSO’, the boundary film formed by neat P-SiSO produced a darker appearance when observed by LOM. The topographical analysis by 3DP shows barely detectable wear along the edge of the ball wear scar, seen in Figure 8b, and insignificant wear on the disc as seen in Figure 8c. In the case of neat P-SiSO, the surface roughness within the worn region of the disc was significantly reduced (~40%) in comparison to the unworn region (Sq = 0.116 µm). The valleys seen in Figure 8d are attributed to the original surface texture, and were not affected by wear and, consequently, the reduction in surface roughness can be attributed to the removal of peaks. As shown in the SEM image Figure 8e, the boundary film was only formed on peaks, and no film was observed in the valley region. EDS analysis in Figure 8f indicates that the boundary film is composed of silicate, similar to the boundary film observed in the case of ‘MAC &P-SiSO’ and again in line with previous studies [96,97].

Comparing neat P-SiSO to ‘MAC &P-SiSO’, the detected characteristic energy peak signal of Si is weaker, and the observed coverage is significantly lower in the case of neat P-SiSO. The fact that a stronger boundary film signal is detected in the case of ‘MAC &P-SiSO’ at 0.4 wt% concentration compared to neat (100 wt%) P-SiSO is attributed to the low amount of wear in the case of neat P-SiSO. A worn surface provides exposure of nascent metal that act as favorable sites of boundary film formation, and consequently, if there is very little wear there will also be very little boundary film.

#### 3.4.2. Influence of Atmosphere on Boundary Film Formation

The tribological performance was clearly affected by the atmospheric conditions, as shown in Figure 5 In order to study the effect of atmospheric conditions on the boundary film formation, surface analysis by the complementary technique described in Figure 6, Figure 7 and Figure 8 was applied and further extended. Figure 9 shows the principle of the extended analysis for ‘MAC &P-SiSO’ under 150 N load conditions. The analyzed region can be seen in Figure 9a–b, with the corresponding EDS spectra in Figure 9c. Each EDS-signal is normalized against the average background radiation as described previously, and the curves are shifted in the vertical scale by 4 units in order to distinguish the separate spectra. Figure 9d is produced by dividing the Film spectra retrieved under different atmospheres with the reference spectrum. The relative counts (divided by the reference and shown in log-scale) shows the detected X-ray photons over the energy spectra covering the elements of interest in the boundary films that were produced. For each condition (AIR, N_2_, CO_2_), several film regions were analyzed and combined in order to produce representative signals of the relative counts over the energy spectrum observed in the boundary film. As can be seen in Figure 9d, the composition of the boundary film appears to be insensitive to the atmospheric conditions in the case of ‘MAC &P-SiSO’ under these conditions.

The boundary film composition was very similar in the case of ‘MAC &P-SiSO’ under all atmospheric conditions at 150 N load. Figure 10 provides information about the composition of the films formed under various atmospheres, along with the spatial coverage of the boundary films as observed by LOM and SEM. It is clear that the boundary film mainly consists of Si, O, and S, with P only detected under CO_2_ atmosphere when there is slightly more wear Figure 10c,g. It has been shown in previous work that phosphorous is activated only at very harsh conditions when using neat P-SiSO [96], and the same appears true in the case of P-SiSO in the additive form (‘MAC &P-SiSO’). As shown in Figure 10h, the boundary film formed by neat P-SiSO also appear to be relatively insensitive to the atmospheric conditions, however, there is one major difference under N_2_, where no boundary film could be detected at all. The N_2_ atmosphere corresponded to the most favorable tribological performance, and it is thus likely that the wear under this condition was too insignificant to provide the nascent metal sites and friction energy required to form a significant solid boundary film. It is possible that adsorption of ions is improved under dry N_2,_ leading to a sufficient layer of adsorbed ions that provide excellent tribological performance under the conditions {150 N/2.4 GPa/N_2_}. As seen by the corresponding LOM image in Figure 10e, there appear to be a boundary film formed in the wear track, as indicated by the black regions in the wear track, but it is likely that the film is too thin to be detectable by the EDS analysis even at the lowest feasible accelerating voltage used (5 kV). Another interesting observation was that the mechanism of boundary film formation under CO_2_ appear significantly different to the film formation under AIR or N_2_. As shown in Figure 10c,f, the carbon dioxide atmosphere significantly increases the coverage of the P-SiSO boundary film, whether observed in LOM or SEM. Formation of boundary film under CO_2_ was also observed in the case of neat MAC (not shown), which had significantly improved tribological performance under CO_2_ atmosphere in comparison to AIR or N_2_. Apparently, at high concentration of CO_2_ at atmospheric pressure, the formation of boundary film is enhanced. In the case of neat MAC, this is clearly beneficial, whereas in the case of neat P-SiSO it slightly shifts the wear mechanism towards reduced friction at the expense of slight wear. This observation is in line with conventional boundary lubrication theory, where excessive boundary film formation can trigger a sacrificial wear mechanism that provides reduced friction at the expense of increased wear [61]. It should also be noted that the Si signal is reduced in the case of neat P-SiSO under CO_2_ conditions, and that there is a significant increase in the carbon signal, which could indicate a change in the composition of the boundary film, and not only the amount of coverage.

#### 3.4.3. Proposed Model of Lubrication by P-SiSO Boundary Film

Based on the surface analysis, a model of lubrication with P-SiSO in neat form or as additive (‘MAC &P-SiSO’) is proposed in Figure 11. In this model, P-SiSO is initially adsorbed onto the lubricated surfaces, providing effective lubrication without significant formation of a solid boundary film, as observed with neat P-SiSO under conditions {150 N/2.4 GPa/N_2_}. The mechanism of lubrication by the adsorbed P-SiSO cannot be conclusively determined from this study, however, three plausible mechanism of lubrication by the adsorbed layer are proposed: (i) adsorbed cations could act analogous to organic friction modifiers, with the phosphonium cation resembling a conventional organic friction modifier [112], as suggested previously [95]. Another possibility (ii) is that the adsorbed layer consists of both anions and cations arranged in ordered layers, providing low friction by interlayer shearing [113]. The third option (iii) is that the adsorbed layer increases the viscosity in comparison to the base fluid, and thereby provides a hydrodynamic lubrication contribution in a manner similar to the mechanism of polymer-based viscous surface films [114].

In any case, if the lubrication conditions become harsher, so that the adsorbed layer is insufficient to protect the steel surfaces, nascent metal is exposed by wear while frictional heating provides activation energy for chemical reactions. At this second step, a silicate boundary film is formed by decomposition of the anion. The silicate film provides effective surface protection under the investigated boundary conditions. Determining the mechanism of friction and wear reduction by the silicate film requires further analysis, however, it is clear that the boundary film provides a smooth surface and that the film prevents adhesion by passivating the nascent metal sites. SEM images indicate that the silicate film is being worn, and it is likely that it is continuously reforming in a manner similar to conventional anti-wear agents [115]. As shown in the proposed model, the silicate layer may also coexist with an adsorbed layer, and in that case the smoothness of the film would be an important factor in maintaining low friction.

The final step of the proposed model is reaction of phosphorous as the phosphonium cation decomposes under harsh tribological conditions. At this step, it is known from previous work [96] that the composition of the boundary film shifts from Si towards P. At the same time, the friction stabilizes at a significantly increased level, and significant wear and increase in surface roughness occur in a manner similar to extreme pressure lubricant agents.

The influence of atmospheric conditions on tribological results and the observed boundary film formation can be explained based on the proposed model. The gaseous atmospheres (N_2_, CO_2_) infer dry conditions (<5%RH), whereas AIR conditions infer exposure to atmospheric humidity. Comparing AIR and N_2_, it is reasonable to assume that the slightly increase friction and wear seen under AIR with neat P-SiSO is due to the adsorption mechanism of P-SiSO being affected by competitive adsorption by H_2_O in the humid environment, which negatively affects the lubrication performance and consequentially leads to increased wear. As nascent metal is exposed by wear, the formation of silicate film increases, which explains the stronger observation of silicate under AIR conditions.

Regarding the influence of CO_2_ atmosphere, the results of neat P-SiSO suggest that friction is reduced under CO_2_, while wear increases slightly compared to AIR or N_2_. The observed silicate boundary film coverage was significantly increased when rubbed under CO_2_ conditions, indicating increased reactivity under CO_2_. From the outgassing results, in Figure 2, a hygroscopic tendency can be seen for neat P-SiSO, meaning that samples exposed to humidity during storage are likely to provide absorbed H_2_O. In the presence of CO_2_, as H_2_O is combined with CO_2_, it is possible that carbonic acid is formed, leading to a consequent increase of boundary film as described in the hypothetical reaction scheme outlined in Equations (7)–(12).
(7)$${\mathrm{CO}}_{2}+\;2H_{2}O⇌\;\left[H_{3}O\right]\left[{\mathrm{HCO}}_{3}\right]$$
(8)$$\left[R_{4}P\right]\left[{\mathrm{TMSC}}_{2}H_{4}{\mathrm{SO}}_{3}\right]\;+\;\left[H_{3}O\right]\left[{\mathrm{HCO}}_{3}\right]\;⇌\;\left[R_{4}P\right]\left[{\mathrm{HCO}}_{3}\right]\;+\;\left[H_{3}O\right]\left[{\mathrm{TMSC}}_{2}H_{4}{\mathrm{SO}}_{3}\right]$$
(9)$$M\to M^{⊕}+e^{⊝}$$
(10)$$\left[H_{3}O\right]\left[{\mathrm{TMSC}}_{2}H_{4}{\mathrm{SO}}_{3}\right]\;+\;M^{⊕}⇌H^{⊕}+H_{2}O+{\mathrm{TMSC}}_{2}H_{4}{\mathrm{SO}}_{3}M\;⇉\;Boundary\;film$$
(11)$$H^{⊕}+e^{⊝}\to H^{\cdot }$$
(12)$${2H}^{\cdot }\to H_{2}$$

The influence of CO_2_ may be of interest for space applications intended for exploration of Mars, where the atmosphere is composed of 95% CO_2_. However, it should be noted that this study was performed at terrestrial atmospheric pressure (101 kPa) instead of Mars atmospheric pressure of <1 kPa, and consequentially the observed influence of CO_2_ is expected to be less pronounced in a Martian environment than in these experiments.

## 4. Conclusions

This work shows the feasibility of employing the ionic liquid P-SiSO as a performance ingredient in multiply alkylated cyclopentane (MAC) space grade lubricants. No attempt was made to optimize the concentration of P-SiSO in MAC, and the lubricant outgassing performance was not adversely affected by the incorporation of 0.4 wt% P-SiSO in MAC, indicating a possibility of adjusting the concentration in order to tune lubricant performance. At 0.4 wt% concentration of P-SiSO in MAC, several effects of interest to space grade lubricants were observed, including improved electric conductivity, reduced friction, and improved anti-wear performance. The main conclusions can be summarized as follows:The addition of 0.4 wt% P-SiSO in MAC did not adversely influence the outgassing performance of the fluid under the conditions of the thermal vacuum outgassing experiments that were conducted in accordance with ECSS-Q-70-02. The outgassing performance of neat P-SiSO indicate that up to 10 wt% P-SiSO could be incorporated in MAC without exceeding the acceptance limits for screening of space grade materials.Incorporating P-SiSO in MAC increased the electric conductivity of the fluid to a level that can be considered relevant for avoiding electrostatic discharge under spacecraft charging conditions (~100 pS/m). The fluid conductivity was attributed to ion mobility, which is time- and temperature-dependent.The tribological performance of the lubricant ‘MAC &P-SiSO’, consisting of 0.4 wt% P-SiSO in MAC was significantly improved in comparison to neat MAC. The effect was observed under high vacuum as well as under atmospheric pressure conditions in air-, nitrogen-, and carbon dioxide-rich atmospheres.The tribological performance was mainly attributed to improved boundary lubricating performance due to ionic adsorption followed by formation of silicate boundary film.The atmospheric conditions have a strong influence on the boundary film formation. Although no major difference was detected in the chemical composition of the boundary film, it was evident that the amount of boundary film coverage was significantly influenced by the atmospheric conditions. Furthermore, under dry nitrogen conditions, effective lubrication by neat P-SiSO was achieved even without evidence of significant reacted boundary film, indicating that ionic adsorption may be an important component of the boundary lubrication performance.

This work shows that ionic liquids of the type P-SiSO display several characteristics that make them useful candidates as performance ingredients in space lubricants. This ionic liquid could also be of interest in terrestrial lubricant applications, where alternatives to the conventional ZDDP triboimprovers are in high demand.

## Figures and Tables

**Figure 1 molecules-26-01013-f001:**
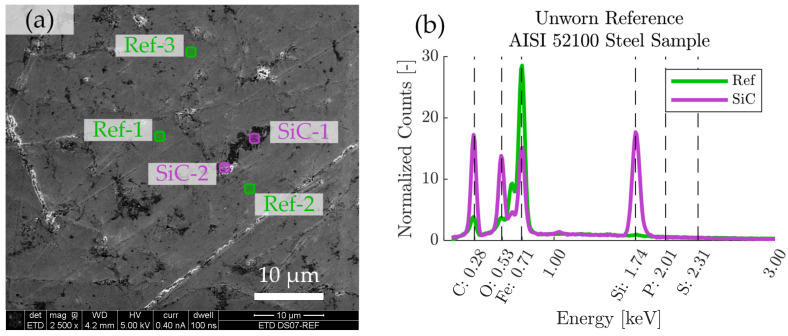
SEM-EDS characterization of reference AISI 52100 steel sample. (**a**) SEM image showing locations of analyzed regions of interest, consisting of three reference regions (Ref-1–3) and two regions rich in silicon-carbon (SiC-1–2). (**b**) Average EDS spectra corresponding to the Ref and SiC regions of interest.

**Figure 2 molecules-26-01013-f002:**
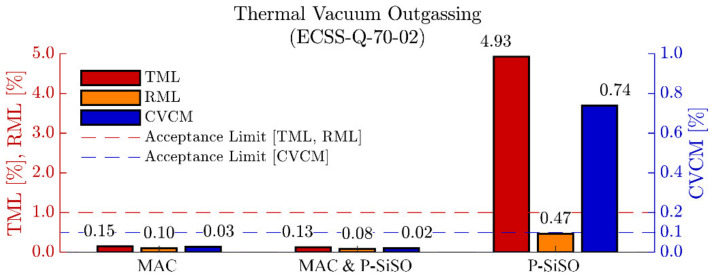
Thermal vacuum (125 °C and <10^−3^ Pa) outgassing test for screening of space materials, in accordance to standard ECSS-Q-70-02. The outgassing performance in terms of total mass loss (TML), relative mass loss (RML), and collected volatiles condensable material (CVCM) is within the acceptance limits for neat MAC, and MAC with 0.4 wt% P-SiSO (MAC and P-SISO). Neat P-SiSO display TML and CVCM outside the acceptance limit, while RML is acceptable.

**Figure 3 molecules-26-01013-f003:**
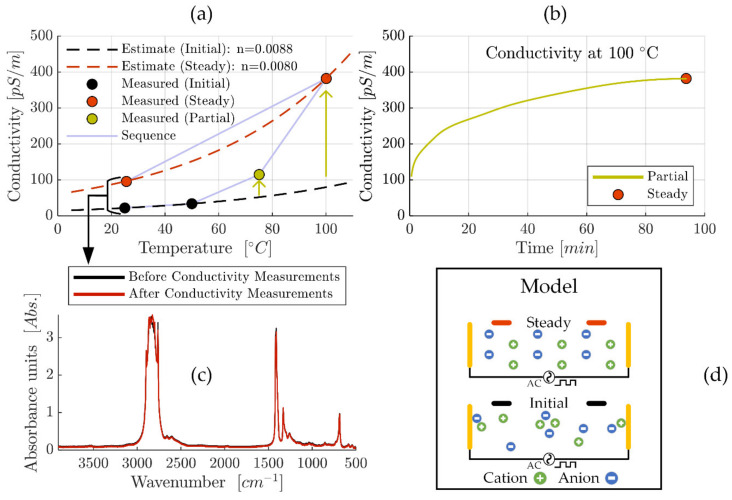
Conductivity analysis of ‘MAC &P-SiSO’, containing 0.4 wt% P-SiSO in MAC. (**a**) Conductivity measured sequentially from 25 °C to 100 °C in increments of 25 °C, before cooling down to 25 °C. Dashed lines represent ASTM D2624 temperature-conductivity relations estimated for initial and steady state. (**b**) Detailed view of initial 100 min of conductivity measurement at 100 °C shows time-dependency of conductivity. (**c**) Fourier transform infrared spectroscopy (FTIR) analysis before first and after last conductivity measurements confirm chemical stability. (**d**) Schematic illustration to explain the observed time-dependency of conductivity.

**Figure 4 molecules-26-01013-f004:**
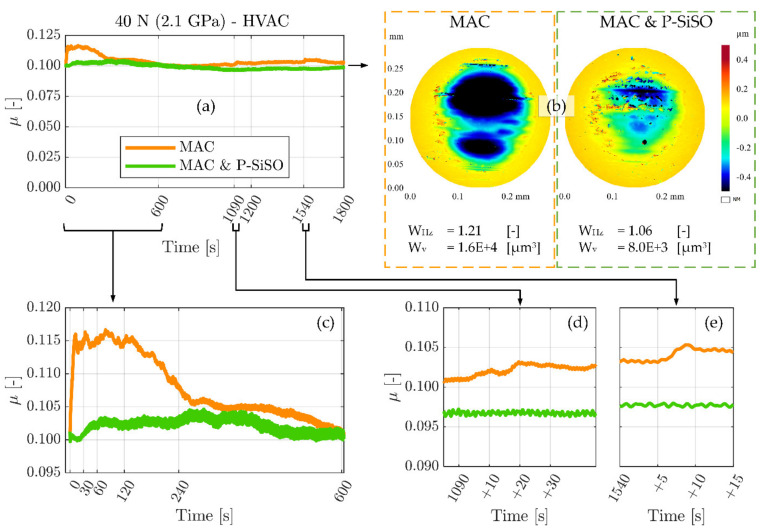
Results from MVT-2 tribotest in high vacuum comparing neat MAC, and 0.4 wt% P-SiSO in MAC, ‘MAC &P-SiSO’. (**a**) Friction traces over 30 min test duration. (**b**) Topographical data (optical 3D-profilometer, 3DP) of the ball wear scars, together with numeric wear indices (W_Hz_) and wear volumes (W_v_). (**c**) Detailed view of the running-in period. (**d**,**e**) detailed views of abrupt friction increase interpreted as indication of partial seizures.

**Figure 5 molecules-26-01013-f005:**
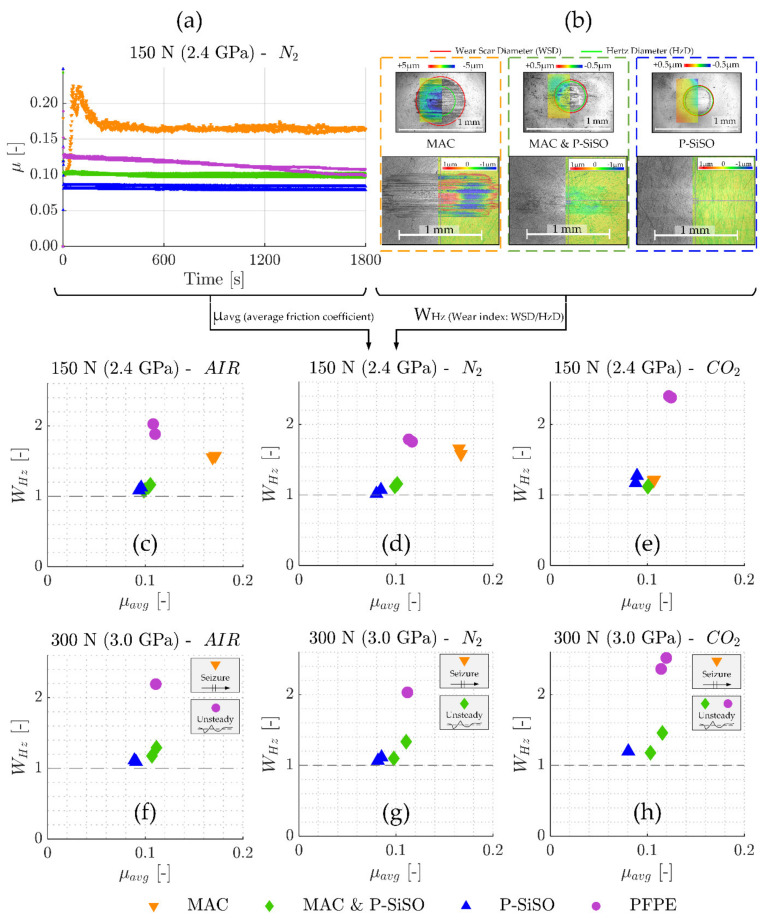
(**a**) Friction traces recorded for neat MAC, ‘MAC &P-SiSO’, neat P-SiSO, and neat PFPE at conditions of {150 N/2.4 GPa/N_2_}. (**b**) Pairs of ball and disc wear scars for neat MAC, ‘MAC &P-SiSO’, and neat P-SiSO, as seen with optical 3D-profilometer (3DP). Color-coded topography data is overlaid on intensity charts to provide representative images of the morphology and severity of wear encountered on both ball (upper part of image) and disc (lower part). (**c**–**h**) Average wear index (W_Hz_) plotted against the average friction coefficient (µ_avg_) at 150 and 300 N load in AIR, N_2_, and CO_2_ atmospheres.

**Figure 6 molecules-26-01013-f006:**
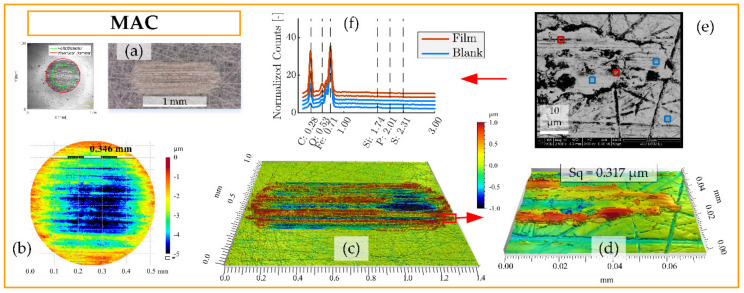
Surface analysis of SRV-3 steel samples lubricated with neat MAC under conditions {150 N/2.4 GPa/AIR}. (**a**) Overview of ball and disc wear scars. (**b**) Detailed view of ball wear scar topography as seen by 3DP with 10× objective (curvature removed). (**c**) Disc wear scar topography seen by 3DP using 10× objective and (**d**) detail at 50× objective. (**e**) scanning electron microscopy with energy-dispersive X-ray spectroscopy (SEM-EDS) analysis of region corresponding to topography in (**d**). (**f**) EDS spectra of regions of interest.

**Figure 7 molecules-26-01013-f007:**
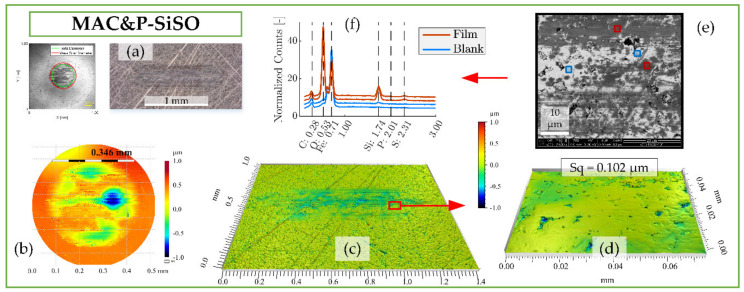
Surface analysis of SRV-3 steel samples lubricated with ‘MAC &P-SiSO’ under conditions {150 N/2.4 GPa/AIR}. (**a**) Overview of ball and disc wear scars. (**b**) Detailed view of ball wear scar topography as seen by 3DP with 10× objective (curvature removed). (**c**) Disc wear scar topography seen by 3DP using 10× objective and (**d**) detail at 50× objective. (**e**) SEM-EDS analysis of region corresponding to topography in (**d**). (**f**) EDS spectra of regions of interest.

**Figure 8 molecules-26-01013-f008:**
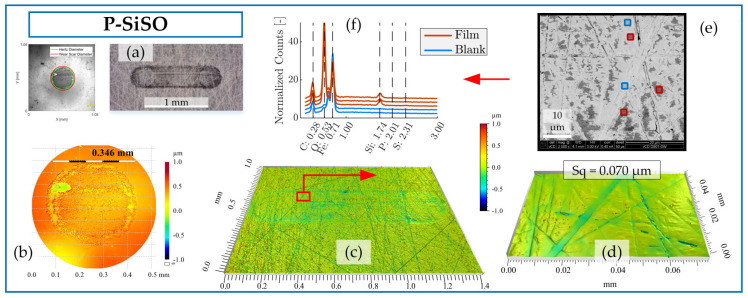
Surface analysis of SRV-3 steel samples lubricated with neat P-SiSO under conditions {150 N/2.4 GPa/AIR}. (**a**) Overview of ball and disc wear scars. (**b**) Detailed view of ball wear scar topography as seen by 3DP with 10× objective (curvature removed). (**c**) Disc wear scar topography seen by 3DP using 10× objective and (**d**) detail at 50× objective. (**e**) SEM-EDS analysis of region corresponding to topography in (**d**). (**f**) EDS spectra of regions of interest.

**Figure 9 molecules-26-01013-f009:**
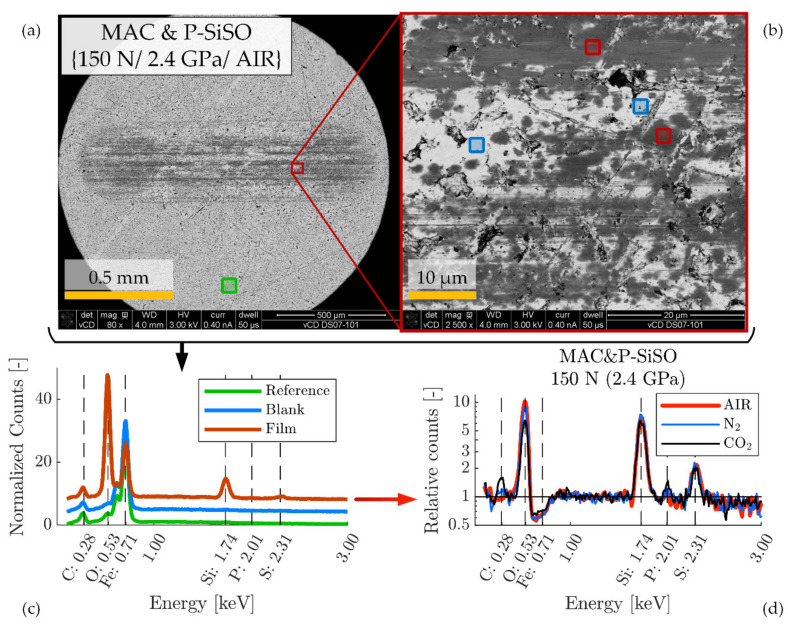
SEM-EDS analysis of SRV-3 steel disc sample lubricated with neat ‘MAC &P-SiSO’ under conditions {150 N/2.4 GPa/AIR}. (**a**) Low magnification SEM image of wear scar with locations of analyzed region and unworn reference. (**b**) High magnification SEM image showing wear scar region analyzed by EDS. (**c**) EDS spectra for unworn region (reference), boundary film region (Film), and worn region without visible film (Blank). (**d**) Relative counts (in relation to reference) in log-scale shows detected X-ray photons over the energy spectra covering the elements of interest in the boundary films that were produced under conditions {150 N/2.4 GPa/AIR}, {150 N/2.4 GPa/N_2_}, {150 N/2.4 GPa/CO_2_}.

**Figure 10 molecules-26-01013-f010:**
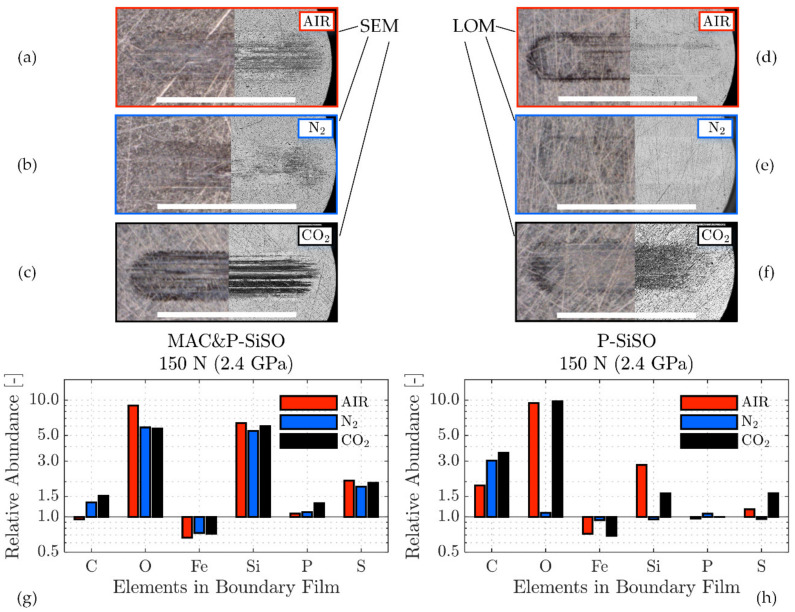
Boundary film analysis of SRV-3 steel discs. (**a**–**f**) Complementary microscopic techniques (left part light optical microscopy (LOM) and right part SEM) indicate the boundary films coverage in the wear scars lubricated with ‘MAC &P-SiSO’ (**a**–**c**) and neat P-SiSO (**d**–**e**) under conditions {150 N/2.4 GPa/AIR}, {150 N/2.4 GPa/N_2_}, {150 N/2.4 GPa/CO_2_}. (**g**–**h**) Relative abundance (in relation to unworn Reference) of elements detected in the boundary films when lubricated with ‘MAC &P-SiSO’ (**g**) or neat P-SiSO (**h**).

**Figure 11 molecules-26-01013-f011:**
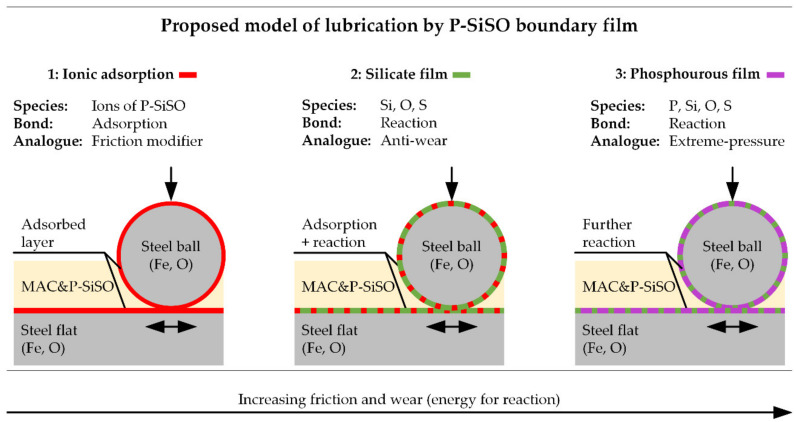
Proposed model of boundary film formation with P-SiSO, based on surface analysis. Ionic adsorption is followed by chemical reactions of silicate and eventually phosphate as the energy available for chemical reactions increases.

**Table 1 molecules-26-01013-t001:** Conditions of tribology experiments.

Tribometer	Gas [−]	F [N]	P_max_ [GPa]	T [°C]	S [m]	Sq [µm]	ΛMAC ^a^ [−]
MVT-2	HVAC	40	2.1	40	90	0.009	0.97
SRV-3	Air/N_2_/CO_2_	150–300	2.4–3.0	25	180	0.061	0.37–0.35 ^b^

^a^ All Λ values calculated using viscosity of multiply alkylated cyclopentanes (MAC) at the test temperature. ^b^ Mean values of Λ over the stroke. The corresponding maximum values of Λ are 0.52–0.50, calculated at maximum speed (midstroke).

## Data Availability

Data is contained within the article or Appendix A.

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
