# Peer review of "Ionic Liquids as Performance Ingredients in Space Lubricants"

_molecules, 2021, doi:10.3390/molecules26041013_

Round 1

Reviewer 1 Report

In the paper “Ionic Liquids as Performance Ingredients in Space Lubricants”, the tribological performance of a hydrocarbon-mimicking ionic liquid dissolved in a multiply alkylated cyclopentane has been studied under vacuum or under N2, CO2 and Air atmospheres.

A very interesting study has been carried out in order to demonstrate the possibility of using this new lubricant in space exploration applications. It is a very high quality work, which includes an extended introduction, conductivity experiments, evaluation of friction and wear, and analysis of worn surfaces.

With respect to the tribological results, not only a significant reduction of the friction coefficient is reached for the new lubricant with respect to neat multiply alkylated cyclopentane, but also models of lubrication by boundary films have been proposed.

I would propose a minor revision before being published:

- From page 9 onwards it is possible to observe error messages in the self-references.

- I would propone to review references in the text to Table 1 in sections 2.4.1 and 2.4.2., maybe rewriting for better understanding.

-I would like to know how many test have been carried out to obtain the results presented in the paper.

- It is necessary to standardize notation for large numbers, or the scientific or the engineering one (e.g. 10-3 Pa in pag 8 and 5E-3 in pag 10).

- In pag. 8, section 2.3 units of “24” do not appear.

-Please, increase the size of Figures 6, 7 and 8.

Author Response

General comment:

In the paper “Ionic Liquids as Performance Ingredients in Space Lubricants”, the tribological performance of a hydrocarbon-mimicking ionic liquid dissolved in a multiply alkylated cyclopentane has been studied under vacuum or under N2, CO2 and Air atmospheres.

A very interesting study has been carried out in order to demonstrate the possibility of using this new lubricant in space exploration applications. It is a very high quality work, which includes an extended introduction, conductivity experiments, evaluation of friction and wear, and analysis of worn surfaces.

With respect to the tribological results, not only a significant reduction of the friction coefficient is reached for the new lubricant with respect to neat multiply alkylated cyclopentane, but also models of lubrication by boundary films have been proposed.

I would propose a minor revision before being published:

General response: We thank the reviewer for reviewing the paper. We agree with the overall interpretation, and are happy that the work is considered of interest and high quality. Overall, we agree that the points raised are valid, and we have made our best effort to address them in order to improve the paper. The specific points and responses are described below.

Point 1: - From page 9 onwards it is possible to observe error messages in the self-references.

Response 1: These errors have been corrected. The links to Table 1, Figure 2, and Figure 11 were unfortunately broken in the original document.

Point 2: - I would propone to review references in the text to Table 1 in sections 2.4.1 and 2.4.2., maybe rewriting for better understanding.

Response 2: This is related to the above point. The broken link caused a duplication of some of the text related to Table 1, and we have now addressed this and carefully checked the links.

Point 3: -I would like to know how many test have been carried out to obtain the results presented in the paper.

Response 3: We have clarified that each test was repeated twice by revising the text in the Method section (2.4.2).

Point 4: - It is necessary to standardize notation for large numbers, or the scientific or the engineering one (e.g. 10-3 Pa in pag 8 and 5E-3 in pag 10).

Response 4: We agree that this should be standardized. We have revised the text (including figure captions) so that the scientific notation is used throughout the document now.

Point 5: - In pag. 8, section 2.3 units of “24” do not appear.

Response 5: Thank you for spotting the missing unit (h), we have corrected this error.

Point 6: -Please, increase the size of Figures 6, 7 and 8.

Response 6: We have increased the fontsize of the smallest axis in Figures 6–8(d). Also, we have provided the figures in higher quality to improve the resolution in order to better resolve the details.   

Reviewer 2 Report

This paper repots on the potential utility of a specially-designed ionic liquids as an additive to lubricants. The ionic liquid design is unique and the obtained results are interesting. The investigation and discussion on the experimental results are well performed. Therefore, I feel this paper is worthy of being published in ”molecules” after minor revision.

One important problem is that there are some phrases such as “in Error! Reference source not found”. Please check these points and revised them.

I feel that the proposed model of the adsorption behavior of ionic liquids in Figure 11 is very reasonable. This explanation is well consistent with experimental results. Considering this proposed model, I feel that the author’s idea on the time-dependence of ionic conductivity might not be correct. My idea is that an adsorption of ionic liquids on electrode surfaces proceeds as time goes by, which leads to the increase of the ionic conductivity at around the electrode surface area. Namely, the increase of local concentration of ionic liquids may result in the increase of the conductivity. My idea is opposite to the author’s idea where the increase of conductivity is attributed to the increase of distribution of ionic liquids. Is it possible to examine the local distribution of ionic liquids by using the SEM-EDS analysis and/or microscopic IR measurements?

Author Response

General comment:

This paper repots on the potential utility of a specially-designed ionic liquids as an additive to lubricants. The ionic liquid design is unique and the obtained results are interesting. The investigation and discussion on the experimental results are well performed. Therefore, I feel this paper is worthy of being published in ”molecules” after minor revision.

General response: We thank the reviewer for reviewing the paper. We are happy that the work is considered of interest and high quality. Overall, we agree that the points raised are valid, and we have made our best effort to address them in order to improve the paper. The specific points and responses are described below.

Point 1: One important problem is that there are some phrases such as “in Error! Reference source not found”. Please check these points and revised them.

Response 1: Indeed, the links to Table 1, Figure 2, and Figure 11 were unfortunately broken in the original document and caused these errors. We have revised this throughout the text. 

Point 2: I feel that the proposed model of the adsorption behavior of ionic liquids in Figure 11 is very reasonable. This explanation is well consistent with experimental results. Considering this proposed model, I feel that the author’s idea on the time-dependence of ionic conductivity might not be correct. My idea is that an adsorption of ionic liquids on electrode surfaces proceeds as time goes by, which leads to the increase of the ionic conductivity at around the electrode surface area. Namely, the increase of local concentration of ionic liquids may result in the increase of the conductivity. My idea is opposite to the author’s idea where the increase of conductivity is attributed to the increase of distribution of ionic liquids.

Response 2: The point raised by the reviewer is important (ion accumulation at electrodes), and the reviewer’s idea of ionic liquid adsorption at electrodes is very reasonable under DC-electric field.

However, in our conductivity experiment, we employed an alternating electric field (AC) as opposed to a unidirectional electric field (DC). In the case of AC-conductivity measurements, the alternating direction of the electric field serves the purpose of inhibiting the accumulation of ions at the electrodes. If we had measured the electric conductivity under DC conditions, we would agree with the reviewer’s model (ion accumulation at electrodes).

We have revised the method description of the conductivity experiment to clarify that we used an alternating current (AC) setup while measuring the conductivity.

Point 3: Is it possible to examine the local distribution of ionic liquids by using the SEM-EDS analysis and/or microscopic IR measurements?

Response 3: To measure the local distribution of ionic liquid within the solvent is certainly an interesting idea, and something that should in principle be possible.

We actually employed FTIR in combination with a sample centrifuge in order to confirm the solubility of the ionic liquid in MAC-solvent. We subjected samples to centrifuging, used pipette to extract top and bottom samples, and measured their FTIR spectra. In principle something similar could be done while the fluid is influenced by an electric field instead of the gravity field. However, we found that the method requires further development before being employed in practice. We think that this part is outside the scope of this investigation, but agree that it is an important point to consider in future work.

Reviewer 3 Report

The authors describe hydrocarbon-mimicking ionic liquids (designated PSiSO) as suitable performance ingredients in multiply alkylated cyclopentane (MAC).

The paper is well written but has to be carefully revised concerning some typing errors essentially and particularly the link to certain references.

The introduction is dedicated to first ionic liquids involving their physical properties and their uses in particular in the lubricant field. This part is well detailed and appropriated with old and recent exhaustive references coming from the authors but also from other teams too.

Next, the part concerning the space grade lubricants is also well described in relation to the tribology. Perhaps at this stage of the MS, I can suggest to reduce this introduction part which is really very long.

The authors described also their ionic liquids, previously prepared and studied, the P-SiSO, which is an association between a tetraalkylphosphonium cation and a trimethylsilylalkylsulfonate anion.

The purpose of the study is then well presented. This work investigates the suitability of employing P-SiSO as an additive in MAC to improve performance in the boundary lubrication process, while simultaneously providing a well-balanced degree of electric conductivity of the lubricant. Concerning the part dedicated to the results, this one is quite complete with outgassing, conductivity, and tribological experiences involving different conditions (essentially quantity of IL as ingredient and various gas environment or vacuum). Light interferometry, followed by scanning electron microscopy with energy dispersive X-ray spectroscopy (SEM-EDS) was also employed for the characterization of boundary films.

In CO2, the authors propose some chemical schemes to explain the boundary films. In Eq. 37, could the authors confirm the presence of H radicals and explain their formation? As well for the proposed model for the lubrication involving phosphate entities, could the authors explain more about this aspect?

No chemical reactions are given for the other atmosphere they have been tested.

The conclusions are also quite complete and long. At the end of these conclusions, the authors specify the detrimental effects on catalytic converters of P-derivatives. My question here is why using such ionic liquids and no other including N-based derivatives or other kind of ionic liquids showing more suitable properties?

In conclusion, in this form, I recommend some major revisions of the MS before a publication in Molecules.

Author Response

General comment:

The authors describe hydrocarbon-mimicking ionic liquids (designated PSiSO) as suitable performance ingredients in multiply alkylated cyclopentane (MAC).

The paper is well written but has to be carefully revised concerning some typing errors essentially and particularly the link to certain references.

General response: We thank the reviewer for reviewing the paper. We are happy that the work is considered in general well written. Overall, we agree with the points raised by the reviewer, and we have made our best effort to address them in order to improve the paper. The specific points and responses are described below.

Regarding the errors associated with links to certain references, we have revised the links to Table 1, Figure 2, and Figure 11 which were unfortunately broken in the original document.

Point 1: The introduction is dedicated to first ionic liquids involving their physical properties and their uses in particular in the lubricant field. This part is well detailed and appropriated with old and recent exhaustive references coming from the authors but also from other teams too.

Next, the part concerning the space grade lubricants is also well described in relation to the tribology. Perhaps at this stage of the MS, I can suggest to reduce this introduction part which is really very long.

The authors described also their ionic liquids, previously prepared and studied, the P-SiSO, which is an association between a tetraalkylphosphonium cation and a trimethylsilylalkylsulfonate anion.

The purpose of the study is then well presented. This work investigates the suitability of employing P-SiSO as an additive in MAC to improve performance in the boundary lubrication process, while simultaneously providing a well-balanced degree of electric conductivity of the lubricant.

Response 1: We do agree that the introduction is long, and we have revised the introduction to improve the readability:

The revised version includes subheadings so that readers can more easily overview the contents of the introduction. We introduced five subheadings: 1.1. Ionic liquid lubricants, 1.2. Outlook on space grade lubricants, 1.3. Considerations in lubricant design, 1.4. Current state of space grade lubricants, 1.5. Contribution of this work.

We believe that this clarifies the structure and improves the readability of the introduction while retaining the high level of detail that we strived for.

Point 2: In CO2, the authors propose some chemical schemes to explain the boundary films. In Eq. 37, could the authors confirm the presence of H radicals and explain their formation?

Response 2: We slightly revised the text describing equations 3.2–3.7 to emphasize that the reaction scheme is hypothetical, and considered under conditions where the tribo-surfaces are rubbed against eachother. We did not attempt to experimentally detect H radicals, but in the reaction scheme we hypothesize that this electrochemical reaction mechanism is possible when the surfaces are rubbed (with consequential emission of electrons as in eq. 3.4).

Point 3: As well for the proposed model for the lubrication involving phosphate entities, could the authors explain more about this aspect?

Response 3: We revised the short description related to formation of the phosphate entities. This part of the model is mostly based on our previous work (ref 96), where we found that the composition of the boundary film changed to one predominately composed of phosphate under the most severe conditions. In the current paper, we again observed a shift in boundary film composition from Si to P under conditions of increased wear. However, in this paper we mostly focus on the first and second steps of the proposed model (adsorption, silicate), and therefore this is reflected in the level of detail in the proposed model.   

Point 4: The conclusions are also quite complete and long. At the end of these conclusions, the authors specify the detrimental effects on catalytic converters of P-derivatives. My question here is why using such ionic liquids and no other including N-based derivatives or other kind of ionic liquids showing more suitable properties?

Response 4: The reviewer raises a very valid point here. There are other ionic liquid alternatives of interest for this specific problem (catalyst poisoning by P), and the problem is complex. We do have some experience with other ionic liquid candidates – including ammonium derivatives (http://dx.doi.org/10.1016/j.triboint.2013.02.020) – and could expand this discussion, however, we prefer to not go into an extended discussion on this topic in this paper in order to avoid diverting the focus from our key points. Therefore, we prefer to revise our final paragraph. In the revised version, we conclude that the investigated type of IL could be of interest as ZDDP alternatives, but we do not go into details. We believe that this conveys the important part of the conclusion, while respecting the complexity of the issue.

Round 2

Reviewer 3 Report

I read the new version of the manuscript. It would have been suitable to mark the revised parts of this one.

The authors replied to most of my comments.

So, I could agree with the publication of this paper in Molecules.